# Tissue Inhibitor of Metalloproteases 3 (TIMP-3): In Vivo Analysis Underpins Its Role as a Master Regulator of Ectodomain Shedding

**DOI:** 10.3390/membranes12020211

**Published:** 2022-02-11

**Authors:** Donatella Pia Spanò, Simone Dario Scilabra

**Affiliations:** Proteomics Group of Fondazione Ri.MED, Research Department IRCCS ISMETT (Istituto Mediterraneo per i Trapianti e Terapie ad Alta Specializzazione), Via E. Tricomi 5, 90145 Palermo, Italy; dspano@fondazionerimed.com

**Keywords:** TIMPs, ADAMs, metalloproteases, ectodomain shedding, proteomics

## Abstract

The proteolytical cleavage of transmembrane proteins with subsequent release of their extracellular domain, so-called ectodomain shedding, is a post-translational modification that plays an essential role in several biological processes, such as cell communication, adhesion and migration. Metalloproteases are major proteases in ectodomain shedding, especially the disintegrin metalloproteases (ADAMs) and the membrane-type matrix metalloproteases (MT-MMPs), which are considered to be canonical sheddases for their membrane-anchored topology and for the large number of proteins that they can release. The unique ability of TIMP-3 to inhibit different families of metalloproteases, including the canonical sheddases (ADAMs and MT-MMPs), renders it a master regulator of ectodomain shedding. This review provides an overview of the different functions of TIMP-3 in health and disease, with a major focus on the functional consequences in vivo related to its ability to control ectodomain shedding. Furthermore, herein we describe a collection of mass spectrometry-based approaches that have been used in recent years to identify new functions of sheddases and TIMP-3. These methods may be used in the future to elucidate the pathological mechanisms triggered by the Sorsby’s fundus dystrophy variants of TIMP-3 or to identify proteins released by less well characterized TIMP-3 target sheddases whose substrate repertoire is still limited, thus providing novel insights into the physiological and pathological functions of the inhibitor.

## 1. Introduction

The human genome encodes for about 15,000 transmembrane proteins, encompassing 19% of the total genome. Ectodomain shedding is a post-translational modification by which membrane proteins are proteolytically cleaved and their ectodomain released. This biochemical mechanism is essential in a number of cellular processes, including cell communication, adhesion and migration. For instance, tissue bioavailability of certain cytokines and growth factors is controlled by ectodomain shedding, as it depends on membrane-level signalling receptors and therefore the ability of cells to respond to specific cues. Moreover, shedding controls cell-to-cell adhesion or adhesion of cells to the ECM, and it modulates cellular transport by controlling levels of endocytic receptors on the cell surface [1]. Members of several different classes of proteases have been shown to function as sheddases, with the metalloproteases playing a most prominent role for the number of substrates that they cleave and the subsequent pathophysiological processes in which they are involved [1,2]. Metalloproteases comprise three related families of proteases, the matrix metalloproteases (MMPs), the disintegrin metalloproteases (ADAMs) and the disintegrin metalloproteases with thrombospondin motifs (ADAMTSs). While most MMPs and ADAMTSs are mainly involved in the catabolism of extracellular matrix components, ADAMs and the membrane-type MMPs (MT-MMPs) act as canonical sheddases [3,4]. Nevertheless, this distinction is not always clear, as membrane-bound ADAMs can be shed themselves and cleave ECM components, and secreted metalloproteases can bind to components of the cell surface and function as sheddases [5,6,7].

Given the large number of substrates and biological processes in which they play a role, the activity of metalloprotease sheddases has to be finely modulated. When dysregulated, aberrant metalloprotease-dependent ectodomain shedding can lead to pathological conditions, such as cancer, arthritis, inflammatory diseases and Alzheimer’s [4,8]. Its regulation occurs, among other mechanisms, by formation of a 1:1 non-covalent complex with the tissue inhibitors of metalloproteases (TIMPs) [9]. Among the four mammalian TIMPs, TIMP-3 has the largest inhibitory profile being able to inhibit not only MMPs and ADAMTSs, but also all the canonical sheddases of the ADAM and MT-MMP families, thus rendering TIMP-3 a master regulator of ectodomain shedding.

This review provides an overview of different functions of TIMP-3 in health and disease related to its ability to control ectodomain shedding. TIMP-3-unique structural features and regulatory mechanisms that enable it to modulate ectodomain shedding, together with in vivo functional consequences of enhanced TIMP-3 expression or ablation, will be discussed. Finally, we describe a collection of mass spectrometry-based approaches that have been used in recent years to systematically identify substrates of TIMP-3 target proteases. These methods have led to the identification of new proteins regulated by the inhibitor. High-resolution proteomics approaches represent a powerful tool to study proteolysis and have provided novel insights into TIMP-3 biology and the functions of its target sheddases, which may be applied in future to other TIMPs or sheddases that have been less well characterized.

## 2. TIMP-3 Structural Determinants Drive Adam Inhibition and ECM Interaction

All mammalian TIMPs appear as “wedge-shaped” with the inhibitory ridge that perfectly complements the active site cleft of target metalloproteases. The TIMP mechanism of inhibition comprises the coordination with the proteolytic Zn^2+^ ion of metalloproteases by their conserved Cys1. This displaces the water molecule needed by metalloproteases for peptide bond hydrolysis [9]. TIMPs consist of two distinct domains, an *n*-terminal domain of about 125 amino acid residues, which retains the full inhibitory capability, and a smaller C-terminal domain whose biological function is not fully elucidated yet, but it has been reported to affect the affinity of TIMPs against ADAMs and therefore ectodomain shedding [10,11]. Despite these common structural features and a similar mechanism of inhibition, the inhibitory profile of the four TIMPs against metalloproteases is different. While other TIMPs mainly inhibit MMPs and have limited inhibitory activities for ADAMs, TIMP-3 can inhibit most ADAMs, including ADAM10, ADAM12, ADAM17, ADAM28 and ADAM33 [9,12]. The majority of interactions between TIMPs and metalloproteases are made by a continuous ridge formed by the *n*-terminal five residues (for instance, Cys1-Thr-Cys-Ser-Pro5 in TIMP-3) that are highly conserved among all TIMPs and residues of the metalloprotease active site cleft [9]. The ability of TIMP-3 to inhibit the greatest number of metalloproteases is due to unique structural features in this region. Furthermore, TIMP-3 is unique in its ability to bind to the ECM, different to other TIMPs that are soluble in the extracellular milieu [9]. Lee and colleagues identified the ECM-binding motif in a cluster of lysine and arginine residues, which is situated on the opposite side of the molecule to the inhibitory ridge [13].

## 3. Regulation of TIMP-3

Given its role in modulating ECM turnover and ectodomain shedding, the bioavailability of TIMP-3 must be finely regulated. This occurs at several levels, including transcriptional regulation, by growth factors and cytokines, and epigenetically by promoter methylation, post-transcriptional regulation by specific miRNAs and by receptor-mediated endocytosis (Figure 1).

### 3.1. Transcriptional Regulation

TGF-β1 is a multifunctional growth factor produced by different cell types, such as monocytes, platelets and chondrocytes, that promotes ECM deposition. It has been shown to induce the expression of TIMP-3 in articular chondrocytes [14]. This occurs by activation of the ERK/MAPK signaling pathway, leading to phosphorylation of the Sp-1 transcription factor and its subsequent binding to the *TIMP3* promoter. Qureshi and colleagues also reported that TIMP-3 expression could be induced by TGF-β1 through different mechanisms involving Smad-2, -3 and -4 and the phosphatidylinositol 3-kinase/Akt signaling pathway [14,15]. Oncostatin M, a cytokine belonging to the IL-6 family, can also induce expression of TIMP-3 in human chondrocytes by activating the Janus Kinase (JAK)/STAT signaling pathway [16]. Epstein–Barr virus (EBV) latent protein 1 (LMP-1) is a viral protein essential for the capability of the Epstein–Barr virus to mediate growth transformation of B-cells. LMP-1 was shown to promote metastasis in EBV-negative nasopharyngeal carcinoma cells by suppressing TIMP-3 transcription [17]. Similarly, human hepatitis B virus suppresses TIMP-3 expression, although the molecular mechanism is not fully elucidated [18]. Missense mutations in the p53 gene inactivate its growth suppressing activity and are observed in the majority of human tumours. Thomas and colleagues showed that missense mutations in p53 created a mutant p53 that gained the ability to bind to the TIMP-3 promoter and suppress its expression [19]. Loss of TIMP-3 expression in cancer cells could lead to an increase of metalloprotease activity and therefore to their increased metastatic potential. Nuclear factor erythroid 2-related factor 2 (Nrf2), a transcription factor with a key role in inflammation, positively regulated the expression of TIMP-3, thereby attenuating inflammatory responses in the liver of mice challenged with ischemia/reperfusion injury (IRI) [20].

### 3.2. Epigenetic Regulation

The epigenetic regulation of TIMP-3 involves a number of mechanisms that are not dependent on its gene sequence. For example, aberrant methylation of the promoter region of TIMP-3 gene was found to be associated with primary cancer of the brain, kidney, colon, breast, etc. [21]. TIMP-3 plays a protective role in cancer, as will be further discussed in the next sections of this review. Epigenetic regulation of the *TIMP3* gene has been reported to be a key mechanism in cancer progression [21,22,23]. Long non-coding RNAs (lncRNAs) are a new class of epigenetic regulators that play important roles in several diseases, including cancer. Long non-coding RNA reprogramming (lncRNA-ROR) was shown to recruit the transmethylase MLL1 to promote H3K4 trimethylation that enhanced TIMP3 transcription [24].

### 3.3. Post-Transcriptional Regulation

Post-transcriptional regulation of TIMP-3 expression has been investigated in both physiological processes and pathological conditions. Several miRNAs have been found to bind the 3′-UTR of the TIMP-3 mRNA, resulting in the degradation of TIMP-3 transcripts and its gene silencing [25,26]. miR-21 is one of the most expressed micro-RNAs in many different tumours, including glioblastoma, and its expression often correlates with tumour grade and poor prognosis [25]. miR-21 targets TIMP-3 mRNAs, leading to their degradation. Its ability to reduce TIMP-3 expression promotes tumour invasion [25,27,28]. Regulation of TIMP-3 by miR-21 has also been observed in physiological processes, including wound healing [29]. During skin wound healing, keratinocytes need to proliferate and migrate. Both keratinocyte proliferation and migration are favored by expression of miR-21, which attenuates TIMP-3 expression [29]. TIMP-3 mRNA is a validated target of miR-181b and miR-206 [26,30]. miR-181b is upregulated by TGF-β and its expression is markedly suppressed in hepatocarcinomas. These data suggest that TGF-β can modulate TIMP-3 expression by miR-181b [30]. miR-206-mediated downregulation of TIMP-3 promotes cardiac regeneration of chronically failing hearts [30]. miR-221 and miR-222 can downregulate the expression of TIMP-3. Garofalo and colleagues found that the hepatocyte growth factor (HGF)/scatter factor upregulates miR-221 and miR-222 expression. This enhances tumourigenicity of lung and liver cancer cells by downregulating the phosphatase and tensin homolog (PTEN) and TIMP3 [31].

Exosomes can be used as means for transferring miRNAs to target recipient cells. Few examples of exosomal miRNAs have been reported to regulate the expression of TIMP-3 and, in turn, cell migration. Melanoma-derived exosomes carried miR21, which promoted invasion of fibroblasts by downregulation of TIMP3 expression [32]. Similarly, macrophage-derived exosomes lowered the expression of TIMP-3 by miR-21-5p and favored ventricular remodeling in mice with myocardial infarction [33].

### 3.4. LRP1-Mediated Endocytosis

Endocytosis has emerged as a major mechanism to regulate bioavailability of TIMP-3 in the tissues, with the low-density lipoprotein receptor-related protein-1 (LRP-1) playing a key role in this process [34,35]. TIMP-3 is constitutively internalized by cells through the receptor, either alone or in a complex with its target metalloproteases, and targeted to lysosomal degradation [34,36]. Shedding of LRP-1 and release of its soluble ectodomain negatively modulate TIMP-3 endocytosis, as soluble LRP-1 is able to bind to the inhibitor and act as a decoy receptor [34,37]. LRP-1 shedding, which is mediated by different metalloproteases, including ADAM17 and ADAM10, can function as a negative feedback loop to regulate ADAM-mediated shedding itself [38,39]. It has been reported that activation of ADAM17 by LPS promotes shedding of LRP-1 and subsequent extracellular accumulation of TIMP-3, which, in turn, dampens ADAM17 activity and TNF release [40]. Thus, LRP-1 shedding and accumulation of TIMP-3 may be a crucial mechanism for the resolution of ADAM-mediated cell responses, including inflammation.

## 4. TIMP-3 Regulates Apoptosis In Vivo

### 4.1. TIMP-3 Is a Pro-Apoptotic Factor in Cancer and Cerebral Ischemia

TIMP-3 has a number of unique features among TIMPs, including the ability to promote apoptosis, as was shown in a number of cancer cells [41,42]. TIMP-3′s pro-apoptotic potential is linked to its ability to stabilize TNF receptors due to inhibition of ADAM17-mediated shedding [43,44]. Bond et al. showed that TIMP-3 is involved in a type II apoptotic pathway initiated by FAS and that a dominant negative form of this death receptor inhibited TIMP-3-induced apoptosis [43]. The ability of TIMP-3 to induce apoptosis has been investigated in vivo in neuronal cell death following cerebral ischemia. TIMP-3 is highly expressed in ischemic neurons and was found to sensitize them to doxorubicin-induced apoptosis [45,46]. *Timp3*-null cortical neurons subjected to oxygen–glucose deprivation (OGD) showed delayed cell death. Loss of TIMP-3 enhanced shedding of FAS ligand and subsequent lower FAS activation and neuronal cell death [46]. As a consequence, *Timp3*-null mice were resistant to middle cerebral artery occlusion (MCAO), indicating that TIMP-3 promotes neuronal cell death in vivo [46]. Additional to neurons, oligodendrocytes (OLs) also undergo a similar TIMP-3-regulated apoptosis during ischemia [47]. Furthermore, TIMP-3 promoted apoptosis in hippocampal neurons upon bilateral carotid artery occlusion (BCAO). Nevertheless, in these cells TIMP-3 may induce apoptosis in a different manner, as *Timp3*-null mice undergoing BCAO had significantly higher levels of shed TNFR1 and ADAM17 activity [48]. Intriguingly, it has also been reported that intracellular TIMP-3 is able to induce apoptosis in synovial and lung fibroblasts by a mechanism that involves NF-kB regulation and which is separate from TIMP-3′s ability to inhibit ectodomain shedding [49].

### 4.2. TIMP-3 Has Anti-Apoptotic Effects in Mammary Gland Involution

The effects of TIMP-3 on apoptosis in vivo were also investigated by Fata and colleagues using mammary gland involution as a model for physiological apoptosis [50]. In response to decreased lactational stimuli, the mammary gland undergoes a series of biological processes leading to its involution, including epithelial apoptosis. Epithelial apoptosis is accelerated in *Timp3*-null mice, and the process is reversed by addition of recombinant TIMP-3, but not TIMP-1. Thus, although the molecular mechanism has not been fully elucidated yet, these data suggest that epithelial apoptosis during mammary gland involution in *Timp3*-null mice is due not to a general lack of MMP inhibition but more likely to one of the unique features of TIMP-3, including its ability to regulate shedding or binding to the ECM [50].

## 5. TIMP-3 Regulates Shedding In Vivo

### 5.1. Human Disorders Associated with TIMP-3

#### 5.1.1. Sorsby’s Fundus Dystrophy

Sorsby’s fundus dystrophy (SFD) is a rare, late-onset macular dystrophy caused by mutations in the *TIMP3* gene [51,52]. SFD is characterized by a pathological deposition of drusen in Bruch’s membrane, the multilaminar basement membrane that supports the retinal pigment epithelium within the macula. This genetic condition, which shares a number of phenotypical similarities with age-related macular degeneration, causes loss of central vision around the fifties [52]. The global structure of TIMP-3 is maintained by six cysteines that are linked by three disulfide bonds in each domain of the protein [53]. All TIMP-3 mutations that are acknowledged to cause SFD introduce an unpaired cysteine residue and promote the formation of higher molecular weight TIMP-3 oligomers: one at the *n*-terminus (S38C) and six at the C-terminus (S156C, G166C, G167C, Y168C, Y172C and S181C) [51,52]. An additional mutation, E139X, impedes the translation of a C-terminal portion of the protein, thus leaving unpaired the cysteine 132 [51]. SFD mutations in TIMP-3 result in an increased accumulation of the protein and a thickening of Bruch’s membrane, leading to reduced permeability for trafficking metabolites and nutrients [54]. However, SFD mutations do not affect the activity of TIMP-3 to inhibit ADAM17-mediated ectodomain shedding, as well as its activity against MMPs and ADAMTSs [55]. Thus, the molecular mechanisms underlying the pathological phenotype promoted by the SFD mutations are not fully elucidated yet, even though the formation of novel disulfide bonds seems to be associated with increased glycosylation of the protein that may change its turnover [51,54].

#### 5.1.2. CADASIL

CADASIL (cerebral autosomal dominant arteriopathy with sub-cortical infarcts and leukoencephalopathy) is an inherited form of cerebrovascular disease that is associated with changes in the number of cysteines causing an odd number of cysteines in the mutant NOTCH3. Such mutations lead to the extracellular deposition of NOTCH3 ectodomain on the vessels. Shed NOTCH3 interacts with TIMP-3 and promotes its accumulation in the vascular extracellular matrix. TIMP-3 plays a crucial role in the pathology of CADASIL. Its haploinsufficiency in a mouse model of CADASIL rescues reduced cerebral blood flow and impaired functional hyperemia, two CADASIL-related phenotypic features [56]. Mechanistically, this occurs by inhibition of ADAM17 and shedding of its substrate heparin-binding EGF-like growth factor (HB-EGF), indicating that TIMP-3 is a major regulator of the ADAM17/HB-EGF/EGFR pathway and cerebrovascular functions [57].

### 5.2. Inflammation

#### 5.2.1. Immune Responses

Tumour necrosis factor alpha (TNF) is an essential pro-inflammatory cytokine in immune responses to invading pathogens. TNF is synthesized as a transmembrane protein that needs to be shed in order to elicit its pro-inflammatory potential. This process occurs by the action of ADAM17 and it is finely regulated by TIMP-3. TIMP-3 deficient macrophages release higher levels of soluble TNF in response to LPS. As a consequence, *Timp3*-null mice display increased susceptibility to LPS-induced septic shock. This phenotype is rescued by either ablation of the TNF receptor 1 (TNFR1) or by ectopic administration of a metalloprotease inhibitor. This study clearly shows that TIMP-3 is a master regulator of inflammation in vivo by controlling the ADAM17/TNF/TNFR1 axis [58]. Other than exhibiting increased inflammation, *Timp3*-null mice failed to resolve the inflammatory process. When subjected to bleomycin-induced lung injury, *Timp3*-null mice showed higher and persistent neutrophilia and macrophage infiltration compared to wild-type mice, mainly due to increased expression of genes associated with proinflammatory (M1) polarization, including Il6, Il12, Nos2 and Ccl2 [59].

Leukocyte migration is a key component of the inflammatory process. Upon pathogen invasion, leukocytes move to sites of infection. This process is mediated by an array of cell adhesion molecules that regulate rolling, adherence of leukocytes to the vascular endothelium and their extravasation out of the circulatory system. Shedding of adhesion molecules is a key mechanism to control leukocyte migration and, as a consequence, TIMP-3 is a major regulator of the process. In fact, TIMP-3 has been reported to reduce shedding of L-selectin, a cell adhesion molecule that is expressed on most circulating leukocytes and plays a major role in directing leukocytes towards the inflammation sites [60]. Its shedding decreases neutrophil rolling velocity in vitro [61], although the same process does not seem to be affected in humans [62]. Similarly, TIMP-3 regulates shedding of intercellular adhesion molecule-1 (ICAM), another key protein involved in leukocyte migration [63]. ICAM is expressed on the endothelial cells and regulates vascular permeability and leukocyte–endothelial cell interactions. Syndecans are a family of four transmembrane proteoglycans. Although they are not canonical adhesion molecules, their role in inflammation has been widely reported, as they can regulate cytokine function and leukocyte extravasation (reviewed in [64]). Syndecans undergo ectodomain shedding upon cleavage of their proteic core and their ectodomains accumulate in biological fluids during inflammatory conditions, such as arthritis [65,66]. Shedding of syndecan 1 and 4, which is regulated by TIMP-3, occurs by the action of ADAM17 in response to pro-inflammatory stimuli [67,68]. Furthermore, syndecan-1 has been reported to be released by the membrane-associated metalloproteinases MT1-MMP and MT3-MMP [69]. Interestingly, syndecans can also be shed by TIMP-3-sensitive soluble metalloproteases. For instance, matrilysin (MMP-7) cleaves syndecan-1 and the gelatinases MMP-2 and MMP-9 can cleave syndecans-1, -2 and -4 [70].

#### 5.2.2. Arthritis

While the ADAM17/TNF/TNFR1 axis plays a key role in defending the organism against invading pathogens, when deregulated, this signalling pathway can lead to autoimmune and inflammatory diseases, including arthritis [58,71,72,73,74]. TIMP-3 plays a protective role in arthritis by inhibiting ADAM17, thereby dampening TNF signalling. This was demonstrated by means of the antigen-induced model of arthritis, which recapitulates some features of human pathology, including inflammation, joint swelling and breakdown of articular cartilage. *Timp3*-null mice subjected to antigen-induced arthritis display a dramatic increase in inflammation and higher levels of circulating TNF compared to wild-type controls [72]. Furthermore, *Timp3*-null mice showed signs of spontaneous cartilage degradation after 6 months of age [75]. In agreement with these studies in mice, it was reported that TIMP-3 inversely correlates with clinical markers and sera levels of soluble CD163, a biomarker of inflammation and macrophage infiltration, in patients with rheumatoid arthritis [76].

#### 5.2.3. Autoimmune Hepatitis

Inflammation is an orchestrated process that involves a heterogenous population of immune and non-immune cells. Murthy and colleagues found that TIMP-3 has a broader role in immunity using a murine model of autoimmune hepatitis in which inflammation is not only driven by canonical immune cells but also epithelial and stromal cells of the liver, such as hepatocytes, endothelial cells and stellate cells [77]. *Timp3*-null mice challenged with intravenous administration of concavalin A to develop autoimmune hepatitis displayed enhanced activation of hepatic T-cells, cytokine release and, ultimately, increased mortality. Mechanistically, loss of TIMP-3 exacerbates the pathology in a TNF-dependent manner, as *Tnf* ablation rescued the increased susceptibility to concavalin A-induced hepatitis. Nevertheless, congenic chimeras generated by transplanting wild-type bone-marrow into *Timp3*-null recipient mice displayed similar pathological features to *Timp3*-null mice, suggesting that TIMP-3 produced by hepatic stromal cells, rather than TIMP-3 released by immune cells, has a prominent role in the pathology of autoimmune hepatitis [77].

#### 5.2.4. Fulminant Hepatitis

Fulminant hepatitis is a rare syndrome leading to rapid liver failure, mostly caused by drug overdose. TIMP-3 was reported to be a major regulator of this process, not only by modulating TNF release and its hepatotoxic consequences but also hepatoprotective signals [78]. Apoptosis induced by the cell death receptor Fas plays a major role in the development of fulminant hepatitis. TNF was found to synergize with Fas agonists, thus increasing Fas-mediated hepatotoxicity. Similarly, shedding of EGFR ligands by ADAM17 sensitizes hepatocytes to FAS-induced apoptosis. On the other hand, TIMP-3 modulates ADAM17-mediated shedding of TNFR1, and its loss had a protective effect, conferring resistance to Fas-induced hepatotoxicity. In conclusion, this model highlighted the complexity of signals that are integrated by TIMP-3 and ADAM17 in vivo, which may promote opposite consequences, such as survival or death upon the induction of hepatotoxic stress [78].

Another model of liver damage, hepatic ischemia/reperfusion injury (IRI), is characterized by elevated ADAM17 activity and TNF levels [79]. *Timp3*-null mice showed exacerbated organ damage and further impaired liver function when challenged with IRI. Intriguingly, additional to an enhanced inflammatory response, loss of TIMP-3 increased ADAM10-dependent shedding of E-cadherin during hepatic IRI. Moreover, a specific ADAM10 inhibitor partly rescued this effect after IRI. Altogether these results indicated that in contrast with the majority of the phenotypes displayed by *Timp3*-null mice, which are related to aberrant activity of ADAM17, some effects of *Timp3* loss in vivo are related to its target protease ADAM10 [80].

#### 5.2.5. Crohn’s Disease

Both ADAM17 and TIMP-3 are constitutively expressed by epithelial cells in the human intestinal barrier, but levels of the inhibitor were found low in patients affected by Crohn’s disease [81]. In order to investigate the role of TIMP-3 in progression of the disease, Monteleone et al. treated *Timp3*-null and Timp3-overexpressing mice with 2, 4, 6-trinitrobenzenesulfonic- acid (TNBS) to induce chronic colitis. They found that *Timp3*-null mice developed severe colitis after administration of TNBS, while mice overexpressing TIMP-3 were resistant to the disease [81]. Although high levels of TNF are associated with Crohn’s disease, ADAM17 seems to play a protective role in ulcerative colitis, mainly for its ability to trigger EGFR signalling and promote epithelial cell growth and goblet cell differentiation.

#### 5.2.6. Atherosclerosis

Atherosclerosis is characterized by chronic inflammation, with macrophages playing a role in all stages of the disease. In order to investigate the role of TIMP-3 in the progression of atherosclerosis, Casagrande and colleagues generated a new mouse model in which TIMP-3 was specifically overexpressed in atherosclerotic plaques via a macrophage-specific promoter (MacT3) and crossed it with an LDLR^-/-^ mouse model of the disease. TIMP-3 overexpression decreased macrophage infiltration and the size of atherosclerotic plaques, indicating a protective role for the inhibitor in the development of atherosclerosis [82]. Conversely, loss of TIMP-3 increased macrophage infiltration and severity of atherosclerotic plaques in the *ApoE*-null mouse model of the disease [83]. Since neither TIMP-1 nor TIMP-2 overexpression fully inhibited atherosclerotic plaque development, the protective role exerted by TIMP-3 may be mechanistically related to its ability to inhibit ADAM17 to regulate TNF release and other key processes in atherosclerosis development [84,85].

### 5.3. Metabolic Diseases

The role of TIMP-3 in controlling metabolism and in the pathogenesis of metabolic disorders has been well documented. *Timp3*-null mice display higher body temperature, oxygen consumption and carbon dioxide production than wild-type mice, regardless of food intake and locomotor activity [86]. This effect on metabolism, caused by TIMP-3 deficiency, is linked with an increase in mitochondrial activity. Obesity-related metabolic disorders, such as insulin resistance, type 2 diabetes mellitus and atherosclerosis, are characterized by low-grade chronic inflammation. Mice with haploinsufficiency of insulin receptor (*Insr*^+/−^ mice) exhibit metabolic dysfunctions of human diabetes, such as glucose intolerance and insulin resistance. *Insr*^+/−^ mice exhibit low levels of TIMP-3. In addition, TIMP-3 expression inversely correlates with the susceptibility of mice to develop diabetes and vascular inflammation [87]. Pharmacological inhibition of ADAM17 led to an amelioration of the pathology, thus indicating that TIMP-3 links the interplay between reduced insulin action and aberrant ADAM17 activity in diabetes and vascular inflammation [87]. A similar correlation between insulin resistance, atherosclerosis and TIMP-3 expression was also observed in humans. TIMP-3 expression was markedly reduced in monocytes from individuals with increased risk of diabetes and atherosclerosis [88]. TIMP-3 function in the development of metabolic disorders associated with obesity was further investigated by crossing the TIMP-3 knockout mouse with the *Insr*^+/−^ mouse. When exposed to a high-fat diet, the resulting *Timp3*^-/-^; *Insr*^+/−^ mouse developed hepatic defects that are features of non-alcoholic fatty liver disease as a consequence of uncontrolled ADAM17 activity [89].

Another pathological consequence of obesity is an increased risk of hepatocellular carcinoma (HCC). As will be discussed in more detail in Section 5.6, TIMP-3 expression negatively correlates with the progression and prognosis of several human cancers. Nevertheless, *Timp3*-null mice are highly resistant to developing HCC in response to a hepatocarcinogen agent (*n*-diethylnitrosamine) [90,91]. The role of TIMP-3 in this process is not linked to its ability to inhibit ADAM17 activity, which was similar in the liver of *Timp3*-null and wild-type mice. Although the mechanism is not yet fully elucidated, *Timp3* ablation led to activation of a number of factors promoting senescence, including p53, p38 and Notch, and to decreased expression of the senescence repressor FoxM1 [90,91]. Thus, these results revealed a new interesting function of TIMP-3 in regulating cell fate.

It has emerged that TIMP-3 can affect gut microbiota and, subsequently, obesity and insulin resistance. *Timp3*-null mice exposed to a high-fat diet show reduced diversity in gut microbiota, liver steatosis and an increase in circulating soluble IL-6 receptors, which, in turn, can activate inflammatory cells to drive metabolic inflammation [92].

### 5.4. Kidneys

TIMP-3 is the most expressed TIMP in the kidneys, and its levels are increased in patients with kidney disease [93]. *Timp3*-null mice subjected to unilateral ureteral obstruction (UUO), a common experimental model of renal injury, developed a more severe hydronephrosis (kidney swelling) compared to wild-type mice [94]. This was a consequence of aberrant tissue thickening. Upon UUO, *Timp3*-null mice displayed higher interstitial fibrosis due to increased fibroblast activation and deposition of type I collagen [93]. Intriguingly, *Timp3*-null mice developed an age-dependent chronic tubulointerstitial fibrosis, even in the absence of renal injury, further underlining a crucial role of TIMP-3 in maintaining kidney function. Renal fibrosis in *Timp3*-null mice is associated with aberrant ADAM17-mediated shedding of TNF, as TNF ablation rescued tissue thinning and the other lesions displayed by *Timp3*-null mice.

Diabetic nephropathy, one of the major complications of diabetes, is characterized by a gradual decline in glomerular filtration rate and progressive albuminuria due to increased accumulation of extracellular matrix and podocyte dysfunction. Loss of TIMP-3 is a hallmark of diabetic nephropathy. Patients affected by the disease show decreased levels of the protein, and TIMP-3 polymorphisms are associated with the disease [95]. Ablation of TIMP-3 in mice exacerbated the pathology in two different models of diabetic nephropathy, the streptozotocin-induced model and the Akita mouse. First, *Timp3*-null mice challenged with streptozocin exhibited increased signs of fibrosis, thicker glomerular basement membrane and albuminuria compared to wild-type mice [96]. Secondly, *Timp3* was genetically ablated in the Akita mouse, which carries an autosomal dominant allele consisting of a missense mutation in the *Ins2* gene and recapitulates aspects of human diabetes [97]. *Timp3* ablation in this mouse exacerbated the diabetic nephrology-like pathology, characterized by increased albuminuria, mesangial matrix expansion and kidney hypertrophy [98]. There is a large amount of evidence for ADAM17 playing a role in the pathogenesis of diabetic nephropathy. Pharmacological inhibition, as well as ADAM17 knockdown, prevents the majority of effects induced by high glucose levels on kidney cells, including NADPH oxidase activity, Nox4 protein and collagen IV expression [99]. Since the progression of diabetic renal injury in *Timp3*-null mice was accompanied by increased ADAM17 and NADPH oxidase activity, it is conceivable that TIMP-3 mainly exerts its function in the kidney by regulating ADAM17-mediated shedding. In agreement, Casagrande and colleagues demonstrated that *Timp3*-overexpression in myeloid cells protected mice from streptozotocin-induced diabetic nephropathy at the same extent as ablation of ADAM17 in podocytes [100]. This evidence further proves that the balance between TIMP-3 and ADAM17 is essential to maintain the physiological function of the kidneys and alterations of such a balance lead to kidney disease.

### 5.5. Heart and Vasculature

#### 5.5.1. Aberrant ECM Remodelling in Timp3-Null Mice

Dilated cardiomyopathy is the most common cause of heart failure. Ectodomain shedding, as well as inflammation and extracellular matrix (ECM) remodelling, play a pivotal role in regulating the homeostasis of the cardiovascular system. When dysregulated, these processes can lead to dilated cardiomyopathy and heart failure. Patients with dilated cardiomyopathy express high levels of the sheddases ADAM10 and ADAM15, which may promote integrin shedding and cause a reduction of cell–matrix interactions, thereby contributing to cardiac dilatation. On the other hand, TIMP-3 levels are reduced in patients with dilated cardiomyopathy, suggesting a protective role for the inhibitor [101]. In agreement, loss of TIMP-3 function in mice triggered spontaneous left ventricular dilatation, cardiomyocyte hypertrophy and contractile dysfunction, resembling the symptoms of human dilated cardiomyopathy [102]. These defects are mainly due to dysregulted ADAM17-mediated shedding of TNF, as genetic ablation of *Tnf* in the same *Timp3*-null mouse ameliorates the pathology and prevents heart failure. TNF is a potent activator of MMP expression. Thus, excess TNF released in the absence of TIMP-3 leads to elevated production of MMPs, especially MMP-8, matrix turnover and ventricular dilation [103].

TIMP-3 plays a key role in another cardiovascular condition characterized by an excess of ECM turnover, the aortic aneurysm. TIMP-3 loss caused augmented remodelling of the abdominal aorta, thereby leading to exacerbated aortic aneurysm, in the angiotensin II-induced murine model of hypertension [104]. In addition, TIMP-3 ablation increased cardiac fibrosis in this model. Interestingly, neither ablation of TIMP-1, TIMP-2 nor TIMP-4 displayed similar increased fibrosis in the same model. This suggests that pathological remodelling of the heart strictly depends on the unique ability of TIMP-3 to inhibit not only MMPs but also ADAM-mediated ectodomain shedding or degradation of ECM components by ADAMTSs [105]. In agreement with this evidence, increased fibrosis is due to increased inflammation and stabilization of matricellular proteins, including SPARC and osteopontin. Finally, TIMP-3 controls cardiovascular homeostasis by regulating neonatal cardiomyocyte proliferation through the EGFR signaling pathway. Deficiency in TIMP-3 increased cardiomyocyte proliferation in cardiomyocytes and neonatal hearts [106].

Hypertension, a major risk factor for cardiovascular disease, is finely regulated by ectodomain shedding (i.e., shedding of angiotensin-converting enzyme type 2 (ACE2)), ECM remodelling and inflammation, suggesting a critical role for TIMP-3 in the process. A protective role for TIMP-3 in hypertension and myocardial vascular remodeling in vivo was demonstrated by exposing *Timp3*-null mice to Nω-nitro-L-arginine methyl ester (L-NAME)-induced hypertension. When challenged with L-NAME, Timp3-null mice showed lower blood pressure, less thickened vascular walls, reduced fibrosis and production of reactive oxygen species in cardiac microvessels [107].

TIMP-3 controls the microvascular endothelial cell barrier function, as was shown in a model of sepsis induced by caecal-ligation and perforation [108]. This method allows the release of faecal material into the peritoneal cavity to generate an exacerbated immune response. *Timp3*-null mice displayed higher pulmonary microvascular leakage after sepsis, leading to tissue edema and multiple organ dysfunction. These defects are associated with dysfunction of microvascular endothelial cells caused by aberrant shedding of cadherins and other proteins that contribute to cell-to-cell adhesion [108].

#### 5.5.2. Unbalanced Angiogenesis in Timp3-Null Mice

TIMP-3 is a potent inhibitor of angiogenesis. This activity is partly separated from metalloprotease inhibition and due to its direct binding to VEGF receptor 2. In turn, the interaction of TIMP-3 with VEGFR2 prevents the binding of VEGF to the receptor and its downstream signaling and angiogenesis [109]. Furthermore, TIMP-3 binds to angiotensin II type-2 receptor, and overexpression of both TIMP-3 and angiotensin II type-2 receptor synergistically inhibits angiogenesis [110]. In agreement, loss of TIMP-3 leads to defective angiogenesis in vivo. *Timp3*-null mice exhibited abnormalities in retinal vasculature, comprising dilated capillaries throughout the choroid [111]. In addition, the *Timp3*-null mouse displayed an increase of VEGF signalling and pathologic angiogenesis in two different models of corneal neovascularization [112]. On the other hand, the ability of TIMP-3 to inhibit ADAM17 was reported to modulate pathological neovascularization, in addition to its ability to control VEGF signalling. *Timp3*-null mice showed greater revascularization compared to littermate controls in a mouse model of oxygen-induced retinopathy. Conversely, pharmacological inhibition of ADAM17 decreased pathological neovascularization [113].

### 5.6. Cancer

TIMP-3 is downregulated in several human cancers, and its loss correlates with worse prognoses [114]. The protective function of TIMP-3 in cancer is complex and exerted by different molecular mechanisms. These anticancer effects include (a) inhibition of cancer cell migration and invasion, mainly by blocking the ability of MMPs to degrade the ECM [115,116]; (b) induction of apoptosis [42,43,44]; (c) inhibition of angiogenesis [117]; (d) improvement of innate immune responses against cancer cells [118]; and (e) inhibition of the epithelial-to-mesenchymal transition [115,119].

While properties of TIMP-3 in cancer have been extensively reviewed elsewhere [120,121], this review focuses on effects of TIMP-3 on ectodomain shedding that have functional consequences in cancer progression. Shedding of adhesion molecules promotes cancer cell migration and invasion and there is growing evidence that inhibition of their shedding by TIMP-3 leads to decreased cancer progression. For instance, TIMP-3 has been shown to block E-cadherin shedding and cancer progression in a mouse model of skin tumour [122]. Shedding of activated leukocyte cell adhesion molecule (ALCAM) has a crucial role in the progression of epithelial ovarian cancer (EOC). TIMP-3 blocks ADAM-mediated ALCAM shedding and EOC migration and invasion [123]. In addition, TIMP-3 loss promotes cancer progression by reducing cell adhesion and increasing beta-catenin signaling [124]. CD44 is a receptor for hyaluronan and plays a central role in cancer by triggering signals that induce cell migration and invasion. Its activity is dampened by a dual shedding mechanism that involves ADAMs, which release its whole ectodomain, or MT1-MMP, which processes CD44 at different sites. TIMP-3 is the only TIMP able to block both shedding mechanisms, thus regulating CD44 signalling in cancer progression [125].

A crucial event in the innate immunity to tumours is the engagement of the NK group 2D (NKG2D) receptor on natural killers (NKs) and NKG2D ligands on tumour cells. Such an engagement overcomes inhibitory signals on NK cells, activates NK cells to release cytotoxic molecules, such as perforin and granzyme, and triggers apoptosis of tumour cells. Thus, proteolytic shedding of ligands for (NKG2D) receptors is a strategy used by tumours to modulate immune recognition by NK cells and cytotoxic T cells. Shedding of NKG2D ligands, including MICA, MICB and ULBP2, is inhibited by TIMP-3 in acute myeloid leukemia (AML) cells, and this process enhances their sensitivity to lytic activity of NK [118].

Some effects promoted by TIMP-3 loss are not evident due to redundant functions of the other TIMPs. This is the case in cancer-associated fibroblast (CAF)-like state regulation. CAFs play a crucial role in cancer progression for their ability to secrete stromal factors that promote cell migration and invasion. Ablation of all four TIMPs induces fibroblasts to acquire a CAF-like state and to secrete exosomes enriched in ADAM10 [126]. This, in turn, stimulates Notch receptor activation in breast cancer cells and enhances motility through the GTPase RhoA. A compound ablation of TIMP-1 and TIMP-3 in the mouse mammary gland triggers a similar Notch activation, leading to mammary stem cell expansion and accelerated gestational differentiation. This phenotype was not exhibited by other single or compound TIMP knockout mice, as expected from the unique ability of TIMP-1 and TIMP-3 to inhibit ADAM10 and Notch activation [126].

### 5.7. Bone and Cartilage

The function of TIMP-3 in bone and cartilage has been characterized in vivo by analysing the skeletal architecture of *Timp3*-null mice and mice overexpressing *Timp3* in chondrocytes. Loss of TIMP-3 compromises tibial bone mass and structure in both cortical and trabecular compartments, with a corresponding increase in the number of osteoclasts [127]. Such defects lead to reduced bone density and thickness and increased cortical bone porosity. As a consequence, bones of *Timp3*-null mice had reduced load and stress capacity [128]. Intriguingly, *Timp3*-overexpression also led to bone abnormalities, including defects in tibial structure [127]. These studies clearly indicate that TIMP-3 levels must be finely balanced to maintain bone homeostasis. Chen and colleagues further demonstrated that TIMP-3 is the most crucial TIMP in maintaining skeletal homeostasis by generating a mouse model in which each TIMP was ablated (called quadruple TIMP-null strain—QT mouse), together with a QT mouse possessing a single *Timp3* allele [129]. *Timp1*^-/-^, *Timp2*^-/-^ or *Timp4*^-/-^ did not show trabecular loss, which is evident in the *Timp3*^-/-^ mouse. QT mice are clearly smaller in size than wild-type mice and display a dramatic reduction of trabeculae in long bones. The authors showed that this phenotype is due to an imbalance between bone deposition and resorption, which is a consequence of unleashed MMP activity and enhanced osteoclast differentiation. Intriguingly, the latter process is mainly driven by aberrant shedding activity of ADAMs in the absence of TIMPs, especially TIMP-3. Unleashed ADAM activity triggers bone resorption by a dual mechanism. First, excess TNF release drives osteoclast differentiation of osteoclast precursors. Secondly, TNF is responsible for Dkk1 overexpression by osteoblasts, which, in turn, negatively regulates Wnt signaling and therefore osteoblast differentiation [129].

### 5.8. Other Functions of TIMP-3 In Vivo

In addition to functions that TIMP-3 exerts in vivo by regulating ectodomain shedding, there are a number of phenotypes developed by the *Timp3*-null mice which are not direct consequences of shedding regulation but mainly consequences of an excess of ECM turnover.

#### 5.8.1. Lungs

*Timp3*-null animals develop spontaneous air space enlargement in the lungs as a consequence of enhanced degradation of collagen in the peribronchiolar space and disorganization of collagen fibrils in the alveolar interstitium, while no increase in inflammation or fibrosis is displayed [130]. In addition, lungs from *Timp3*-null mice showed decreased bronchiole branching, enhanced activity of MMPs and fibronectin degradation throughout lung development compared to controls [131].

#### 5.8.2. Brain

Extracellular matrix (ECM) turnover, especially degradation of proteoglycans, plays a crucial role in regulating synaptic activity, nerve recovery and cognition. *Timp3*-null mice exhibited deterioration in cognitive function and decreased habituation in the open-field test, as a consequence of enhanced hippocampal MMP activity [132]. Other than ECM proteoglycans, cell surface proteoglycans play a key role in neurite elongation, neuronal plasticity and synapse formation [133]. Neuroglycan C (NGC) is a transmembrane-type of chondroitin sulfate proteoglycan that is exclusively expressed in the central nervous system. The ectodomain of NGC promotes neurite outgrowth and therefore its shedding is a crucial event in this process. This occurs by MT5-MMP, a TIMP-3 target metalloprotease [134]. IgLONs are members of the immunoglobulin superfamily of cell adhesion proteins that negatively regulate neuronal outgrowth. Two members of this family, neurotrimin (NTM) and limbic system-associated membrane protein (LSAMP) can be shed by ADAM10. Thus, TIMP-3 controls neurite outgrowth by regulating shedding of these two proteins [135].

In conclusion, TIMP-3 regulates shedding of numerous proteins, spanning from pro-inflammatory cytokines, such as TNFα, to growth factors and adhesion molecules, thereby modulating a number of biological processes in vivo (Table 1).

**Table 1 membranes-12-00211-t001:** List of transmembrane proteins whose shedding is regulated by TIMP-3.

Substrate	Validation Method	Effects of TIMP-3 on Shedding Inhibition	References
TNFα	ELISA	TIMP-3^-/-^ macrophages release more TNFα in response to LPS than wild-type macrophages–TIMP-3 regulates the ADAM17/TNF/TNFR1 axis	[58]
ALCAM	Western blot	Inhibition of ADAM17-mediated ALCAM release and epithelial ovarian cancer (EOC) cell motility in a wound-healing assay	[123]
CD44	Two-step sandwich enzyme immunoassay (EIA) system	ADAM- or MT1-MMP-dependent shedding of CD44 is inhibited by TIMP-3 in A375 human melanoma cells	[125]
L-selectin	Flow cytometry; ELISA	ADAM17-mediated L-selectin shedding is inhibited by TIMP-3 in mouse and human lymphocytes and monocytes; migration across endothelial monolayers is not affected by the inhibitor	[60]
ICAM-1	Western blot	TIMP-3 regulates ADAM17-mediated ICAM-1 shedding in human kidney fibroblast 293 cells	[63]
Amphiregulin	ELISA	TIMP-3^-/-^ hepatocytes showed enhanced ADAM17-dependent release of amphiregulin and EGFR signaling	[78,136]
HB-EGF	ELISA	Loss of TIMP-3 enhanced ADAM17-dependent release of HB-EGF and EGFR signaling in hepatocytes	[78]
TGFα	ELISA	TIMP-3^-/-^ hepatocytes showed enhanced ADAM17-dependent release of TGF-α and EGFR signaling	[78]
MIC-A	ELISA	TIMP-3 inhibits ADAM17-dependent shedding of MIC-A and enhances lytic activity of NK cells	[118]
MIC-B	ELISA	TIMP-3 inhibits ADAM17-dependent shedding of MIC-A and enhances lytic activity of NK cells	[118]
LRP-1	Western blot	TIMP-3 controls metalloprotease-dependent LRP-1 shedding (ADAM17, ADAM10, MT1-MMP) and therefore LRP-1-mediated endocytosis	[38,39]
CD163	Flow cytometry; ELISA	TIMP-3 inhibited shedding of CD163, an RA biomarker	[76]
TNF-R1	ELISA	TIMP-3 controls ADAM17-dependent TNFR1 shedding, the ADAM17/TNF/TNFR1 axis and systemic inflammation	[58]
TNF-R2	ELISA	TIMP-3 controls TNFR2 shedding and inflammation	[58]
APP	Mass spectrometry; western blot	TIMP-3 inhibited α-secretase cleavage of APP and increased levels of TIMP-3 in AD may contribute to higher levels of Aβ	[137]
APLP2	Mass spectrometry	TIMP-3 inhibited metalloprotease-dependent shedding (probably ADAM10) of APLP2 in HEKs	[138]
Syndecan-1	Dot immunoassay	Ectodomain shedding of Syndecan-1 is specifically inhibited by TIMP-3 in murine and human nonadherent cell lines	[67]
Syndecan-4	Dot immunoassay	Ectodomain shedding of Syndecan-4 is specifically inhibited by TIMP-3 in murine and human nonadherent cell lines	[67]
Ephrin B2	Mass spectrometry	TIMP-3 inhibited metalloprotease-dependent shedding (probably ADAM10) of EphB2 in HEKs	[138]
PTPRK	Mass spectrometry	TIMP-3 inhibited metalloprotease-dependent shedding (probably ADAM10) of PTPRK in HEKs	[138]
Ephrin type-A receptor 4	Mass spectrometry; western blot	TIMP-3 inhibited ADAM10-dependent shedding of EphA4 in HEKs	[138]
CADM-1	Mass spectrometry	TIMP-3 inhibited metalloprotease-dependent shedding (probably ADAM10) of CADM-1 in HEKs	[138]
NEO-1	Mass spectrometry	TIMP-3 inhibited metalloprotease-dependent shedding (probably ADAM10) of NEO-1 in HEKs	[138]
NRP-1	Mass spectrometry	TIMP-3 inhibited metalloprotease-dependent shedding (probably ADAM10) of NRP-1 in HEKs	[138]
PTK7	Mass spectrometry; western blot	TIMP-3 inhibited metalloprotease-dependent shedding (probably ADAM10) of PTK7 in HEK	[139]
E-cadherin	Western blot	TIMP-3 loss induced ADAM10-mediated E-cadherin shedding in hepatocytes and promoted cell death upon liver ischemia/reperfusion injury	[80]
Ephrin B4	AP cell-based assay; western blot	TIMP-3 inhibits ADAM9-mediated shedding of EphB4 in mEFs	[140]
LSAMP	Outgrowth assay	TIMP-3 inhibits ADAM10-dependent shedding of LSAMP and reduces neurite outgrowth from DRG neurons	[135]
NRG-1	Western blot	TIMP-3 downregulation by diet and exercise increased NRG-1 cleavage in vivo	[141]

## 6. Mass Spectrometry-Based Approaches to Investigate Functions of TIMP-3

The function of a protease strictly depends on the collection of substrates that it cleaves. Some proteases cleave a large number of substrates, while others have a very restricted activity towards a few proteins. Furthermore, the activity of proteases can be spatially and/or temporally confined, so that even proteases with a large spectrum of substrates may only access a limited number of them at specific locations and at specific times. In the last years, high-resolution mass spectrometry techniques have been developed to investigate the function of specific proteases by the systematic identification of their substrates in a specific cellular context [2,142]. These proteomics-based approaches have also been applied to TIMP-3, enabling the discovery of new functions of the inhibitor on the stabilization of transmembrane proteins and endocytosis of secreted proteins [138,139]. Here, we will provide a brief description of these mass spectrometry-based techniques that have been used to characterize the function of TIMP-3 and its target proteases and which may eventually be applied to other proteases and their inhibitors.

### 6.1. High-Resolution Secretome Analysis

High-resolution secretome analysis allows the determination of alterations of protein levels in the secretome of cultured cells. This includes the possibility to evaluate levels of proteins that are released by ectodomain shedding, whose levels are reduced when their sheddases are inhibited or genetically ablated (Figure 2). Conditioned media of specific cells (for instance, cells overexpressing TIMP-3) are applied to tryptic digestion, which transforms a complex protein sample containing secreted and shed proteins into a mixture of peptides that can be analysed by mass spectrometry. These peptides are usually separated by C18-reversed phase liquid chromatography (LC) prior to MS/MS analysis. MS/MS allows the search for each peptide against a protein database so that it can be linked to a protein and its relative quantification, for instance, by a label-free quantification method [142].

Secretome of TIMP-3-overexpressing HEK293 cells were analysed using a similar approach and revealed that TIMP-3 mainly inhibited shedding of ADAM10 substrates in these cells and under steady-state conditions [138]. A similar effect on ADAM10 shedding was noted when cells were treated with a molecule able to prevent TIMP-3 endocytosis and increase its extracellular levels [37]. These results complemented the phenotypical analysis of *Timp3*-null mice, whose most prominent phenotypes are related to a deregulated ADAM17 activation [9,105]. Altogether, proteomics and in vivo analysis suggest a dual role for TIMP-3 in regulating the homeostatic turnover of membrane proteins by ADAM10, and several biological processes, including cell-to-cell communication and adhesion, by controlling the stimulated activity of ADAM17.

### 6.2. Surfaceomics

As extensively discussed within this review, shedding regulates levels of transmembrane proteins and therefore biological responses. In turn, TIMP-3 has been reported to stabilize membrane proteins by inhibiting ectodomain shedding, such as death receptors and adhesion molecules, thereby modulating cellular processes. However, shedding can be a means to control the homeostatic turnover of transmembrane proteins. The effects of TIMP-3 on proteins that are shed at a low rate, as a consequence of their constitutive turnover at the cell surface, may be less pronounced and not lead to an evident increase of their levels at the cell surface and clear functional consequences on cell function. In order to discriminate which proteins are effectively stabilized on the cell surface by TIMP-3 (as well as by genetic ablation of specific sheddases of their pharmacological inhibition), quantitative proteomics can be used. Different approaches can be used to isolate cell surface proteins prior to being analysed by mass spectrometry, including enrichment of glycosylated membrane proteins by SPECS (discussed in the next section) and biotinylation of membrane proteins followed by streptavidin pulldown. The latter approach was used to identify proteins stabilized by TIMP-3 at the surface of HEK293 cells [139]. In this study, Carreca et al. demonstrated that only a group of shed proteins effectively accumulate on the cell surface when TIMP-3 is overexpressed and their shedding is inhibited. Such a proteomic approach can complement secretome analysis and may be useful to assess whether shedding of specific proteins can lead to functional consequences for cell behavior.

### 6.3. SPECS and surSPECS

Secretome protein enrichment with click sugars (SPECS) is a proteomic method that has been proven useful to investigate shedding, especially in those cells that require serum and other additives to grow, such as primary neurons [143,144]. SPECS consists of metabolic labelling of cellular glycoproteins with azido sugars, followed by copper-free click chemistry-mediated biotinylation. This process allows one to label and purify only shed proteins, but not serum glycoproteins. Labelled ectodomains released by shedding will be pulled down from conditioned media using a streptavidin-conjugated resin and analysed by high-resolution mass spectrometry. Dibenzocyclooctyne (DBCO)-NHS is usually used as the click linker for biotin-conjugation in SPECS. This compound is not cell permeable, and therefore labelled ectodomains of cell surface transmembrane proteins can be biotinylated by click-chemistry and enriched by streptavidin pulldown in a similar manner to soluble ectodomains [145]. This method, namely, “surface-spanning protein enrichment with click sugars” (SUSPECS), in which the enrichment of cell membrane proteins by click chemistry is followed by mass spectrometry analysis, has been successfully used to investigate ectodomain shedding in neurons.

### 6.4. Tails

Additional to ADAM-mediated ectodomain shedding, TIMP-3 modulates the activity of secreted metalloproteases, such as MMPs and ADAMTSs. These proteases are majorly involved in processing components of the ECM. The previously described proteomics methods, which have been used to investigate ectodomain shedding, are not suitable to identify soluble substrates of TIMP-3 target metalloproteases and therefore they are not appropriate to study the role of TIMP-3 in ECM turnover. In fact, these methods comprise a tryptic digestion of protein samples and evaluation of relative protein abundance. Relative abundance of an ECM component would not change upon cleavage by a protease, and therefore methods that are specifically targeted to identifying cleavages should be used. For instance, the terminal amine-based isotope labeling of substrates (TAILS) has been extensively characterized to investigate MMPs and their inhibitors ([146,147] and reviewed in [148]). Based on the TAILS method, primary amines of both the *n*-termini and lysine residues of proteins are chemically labeled with formaldehyde and isotopes are labeled for the following mass spectrometry analysis. Then, labeled proteins are digested by trypsin, which will cleave after arginines, but it will not be able to cleave after lysines, as they will be blocked after formaldehyde conjugation. Then, TAILS peptides are enriched and subjected to liquid chromatography and high-resolution mass spectrometry. The neo-*n*-termini peptides arising from cleavages of the protease of interest will appear with higher relative abundance, and from this the substrates of specific proteases can be deduced.

## 7. Conclusions and Perspectives

TIMP-3 is an endogenous inhibitor of metalloproteases that is able to inhibit all canonical sheddases of the ADAM family, in addition to MMPs and ADAMTSs. For this reason, TIMP-3 is considered a master regulator of ectodomain shedding. Its genetic ablation leads to several abnormalities related to unregulated shedding, while its overexpression has been proven as protective in a number of pathological conditions characterized by enhanced proteolysis, including cancer and arthritis. Functions of TIMP-3 have been characterized in a targeted manner, by analysis of phenotypes arising in mouse as a consequence of its genetic ablation. Nevertheless, some phenotypes become evident only when mice are challenged with specific stimuli, and bottom-up approaches may be required to uncover such functions of the inhibitor and its target proteases that do not lead to evident abnormalities in vivo under steady-state conditions. In the last few years, unbiased high-resolution proteomics has allowed the systematic identification of protease substrates, thereby providing key information to deduce new functions of proteases and their inhibitors, including TIMP-3. Indeed, proteomics has been used to identify transmembrane proteins that are regulated by TIMP-3 and revealed new molecular mechanisms in which TIMP-3 is involved.

Mutations in the *TIMP3* gene lead to pathological variants of the inhibitor associated with the onset of Sorsby’s fundus dystrophy (SFD), a degenerative disease of the macula. Despite scientific efforts that have been made to understand how SFD mutations affect TIMP-3 function and cell behavior, the molecular mechanisms linking SFD mutations to the onset of the disease are still unknown [51]. The proteomic approaches that we have described in this review may be used to evaluate effects of SFD TIMP-3 mutants on cell behavior that may be linked to the new pathological functions of the inhibitor. Interestingly, all mutations leading to development of SFD occur at the C-terminal domain of TIMP3, highlighting its involvement in pathophysiological processes that are yet to be elucidated. The C-terminal domain of TIMP-3 is dispensable for metalloprotease inhibition in most cases [9]. Nevertheless, similar to the SFD mutations, a truncated form of TIMP-3 lacking the C-terminal domain leads to different inflammatory responses than full-length TIMP-3 after myocardial infarction [51,149]. Methods of high-resolution proteomics, similar to those reviewed in this article, may serve for further characterization of the C-terminal domain of TIMP-3. Furthermore, while the substrate repertoire of ADAM10 and ADAM17 has been extensively characterized, the number of identified substrates of other TIMP-3 target sheddases, including ADAM12 and ADAM15, is still limited [8]. The function of such sheddases strictly depends on the array of their substrates. Thus, the identification of substrates may link the activity of sheddases and their inhibitors, including TIMP-3, to specific biological processes.

In conclusion, TIMP-3 is a major regulator of ectodomain shedding in vivo. Its ablation or overexpression in mice has led to the identification of several biological processes in which TIMP-3 plays a key role. Nevertheless, many aspects of TIMP-3 biology are not fully characterized yet. Proteomics represents a promising tool to uncover such TIMP-3 functions that have not been elucidated by in vivo analysis and targeted biochemical approaches, thus providing new insights into the biology of the inhibitor.

## Figures and Tables

**Figure 1 membranes-12-00211-f001:**
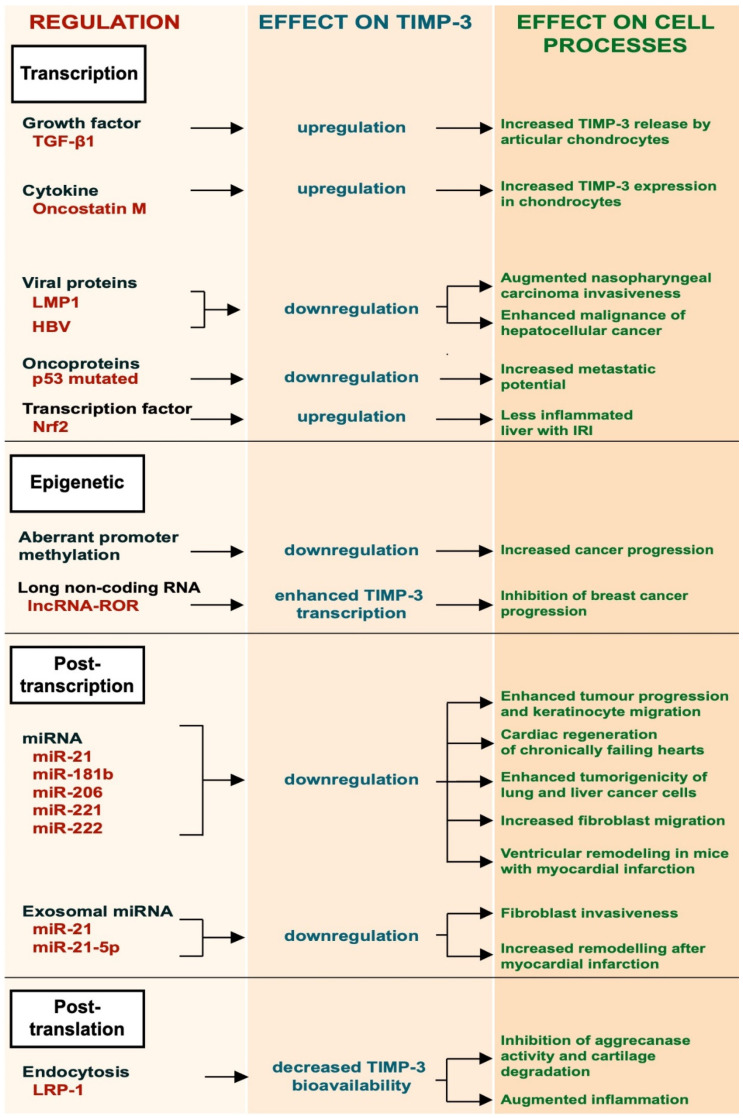
Schematic representation of the different regulatory mechanisms of TIMP-3.

**Figure 2 membranes-12-00211-f002:**
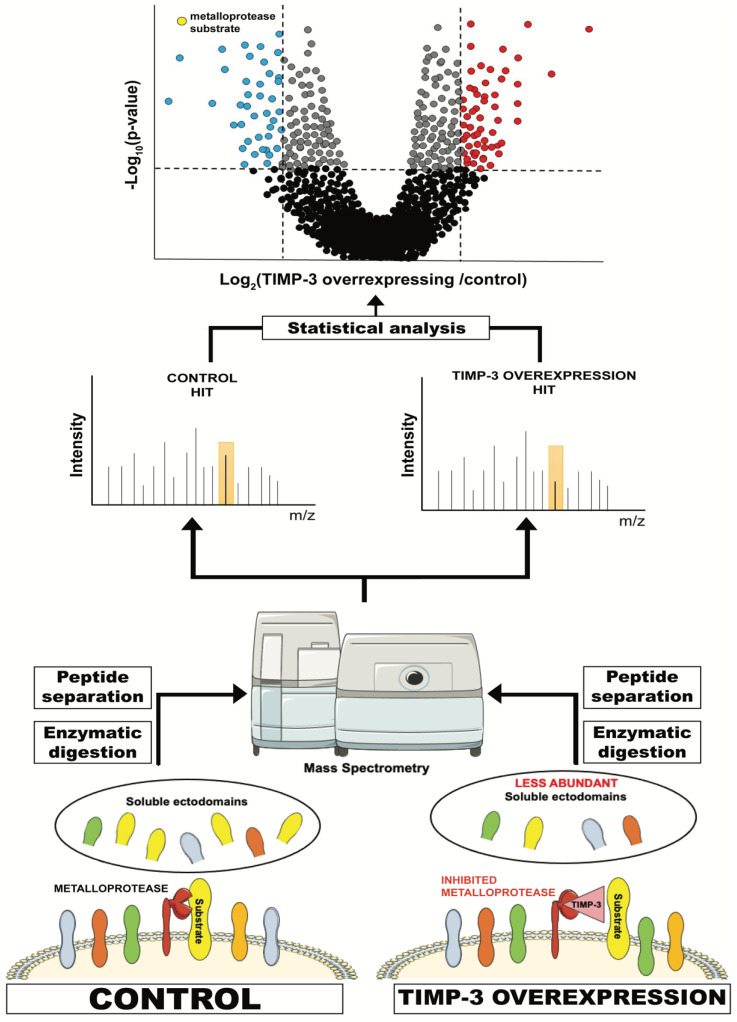
Schematic representation of a typical proteomic workflow to investigate ectodomain shedding. For instance, TIMP-3-overexpressing or control cells are cultured in serum-free media and conditioned media are collected. These contain, among other proteins secreted by the cell, the ectodomain of proteins that are shed by TIMP-3 target metalloproteases. Levels of these proteins in TIMP-3-overexpressing cells will be lower due to inhibition of their shedding by TIMP-3. Conditioned media from TIMP-3-overexpressing and control cells will be applied to tryptic digestion, C18 reversed phase liquid chromatography (LC) and, ultimately, MS/MS analysis. This will enable the identification of proteins contained in the conditioned media and the quantification of their levels in TIMP-overexpressing cells versus controls. Finally, a statistical analysis will show the levels of the proteins that are altered in the media of TIMP-3-overexpressing cells.

## Data Availability

Not applicable.

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
