# Peer review of "Tissue Inhibitor of Metalloproteases 3 (TIMP-3): In Vivo Analysis Underpins Its Role as a Master Regulator of Ectodomain Shedding"

_membranes, 2022, doi:10.3390/membranes12020211_

Round 1

Reviewer 1 Report

The research question is interesting, and the authors have done an excellent job of compiling the review. However, I have some criticism and suggestions before the review to be considered.

  • Rewrite the introduction to retain the emphasis on proteomics tools as well, since the title describes the impartial use of proteomics for the subject matter. Therefore rewrite the section to keep the focus on the topic matter.
  • Sections 2 and 3 aren't very useful for the review since they're too long. The material is outdated and heavily generalized, for which a great deal of information is already accessible in the term, as well as in the review article. So either reduce the amount of material and move it to the introductory section or remove it entirely for more clarity. It is usually preferable for a decent review to remain focused on the subject matter.
  • Please provide the schematic images based on the literature to detail the regulation of TIMP3 based on transcriptional, epigenetic, post-transcriptional, and endocytosis. Club all the points and create the schematics pathway for the activity. Authors could leave the missing link as a question for further research questions.
  • Please include the comprehensive table for all the studies performed so far. In addition, detail the significant finding and type of proteomics tools used.
  • The authors stated the use of proteomics research in the title; nevertheless, only a few instances of the proteomics approach employed in the experiment are found in sections 5 and 6. The preceding sections serve as the overarching explanation of TIMP3. Maintain your attention on the title and engage in discussion with those research that has relevance to the proteomics findings.
  • Please list the ten most essential questions for future study that can be answered based on the research that has been examined.
  • Please add more recent five years' findings and thoroughly check the English and grammar throughout the text.

Author Response

Dear reviewer 1,

Thanks for contributing to improve the quality of our manuscript. We addressed your concerns and followed your suggestions as it is detailed below:

  • Rewrite the introduction to retain the emphasis on proteomics tools as well, since the title describes the impartial use of proteomics for the subject matter. Therefore rewrite the section to keep the focus on the topic matter.
  • We added lines 57-67, in which we introduced the proteomic methods that are described within the manuscript. We hope that the introduction reads now in a thorough manner. 
  • Sections 2 and 3 aren't very useful for the review since they're too long. The material is outdated and heavily generalized, for which a great deal of information is already accessible in the term, as well as in the review article. So either reduce the amount of material and move it to the introductory section or remove it entirely for more clarity. It is usually preferable for a decent review to remain focused on the subject matter.
  • We shortened sections 2 and 3, leaving in all info that is strictly necessary for reading and understanding the rest of the manuscript. We hope that the manuscript now reads with a better flow.
  • Please provide the schematic images based on the literature to detail the regulation of TIMP3 based on transcriptional, epigenetic, post-transcriptional, and endocytosis. Club all the points and create the schematics pathway for the activity. Authors could leave the missing link as a question for further research questions.
  • According to your suggestion, we have added Figure 2, which lists all the regulatory mechanisms of TIMP-3, together with their effect on cell behavior.
  • Please include the comprehensive table for all the studies performed so far. In addition, detail the significant finding and type of proteomics tools used.
  • We believe that table 1, which shows all proteins whose shedding is regulated by TIMP-3, is quite comprehensive of the biological processes that are regulated by TIMP3. We have not added a table to summarize the proteomics results as only two articles have been published so far on this topic. Nevertheless, we tried to stress the point that the proteomics methods that have been applied to TIMP-3 and are described in this article may be used in future to further characterize TIMP-3 biology. These methods may shed light on other processes regulated by the inhibitor and that are not fully elucidated yet (for instance, we added the lines 808-835).
  • The authors stated the use of proteomics research in the title; nevertheless, only a few instances of the proteomics approach employed in the experiment are found in sections 5 and 6. The preceding sections serve as the overarching explanation of TIMP3. Maintain your attention on the title and engage in discussion with those research that has relevance to the proteomics findings.
  • We agree with the reviewer that the proteomic approach to investigate TIMP-3 function is a major topic of this manuscript, and therefore it is mentioned in the title. Nevertheless, proteomics has only been used recently, while in vivo analysis (another main topic as stated in the title) has been largely used to characterize the function of the inhibitor. As a consequence, the large majority of discoveries about TIMP-3 function has been accomplished by ablation or overexpression of TIMP-3 in mouse, and by targeted biochemical approaches. For this reason, the in vivo analysis section and the proteomics section cannot be perfectly balanced in the text. In addition to what has been done so far, we believe that the proteomic approach could be a major way to investigate function of sheddases and inhibitors in the future, and this is why we stressed it in the title and text. We hope that this point is clearer in the revised version of the manuscript, as we discussed it further in the Intro and Conclusions (lines 57-67 and 808-835)
  • Please list the ten most essential questions for future study that can be answered based on the research that has been examined.
  • Following your suggestion, we provided examples of what the proteomic approach could address in terms of TIMP-3 functions that are not fully elucidated yet (lines 808-835). 
  • Please add more recent five years' findings and thoroughly check the English and grammar throughout the text.
  • We added more recent findings, especially in the TIMP-3 regulation section (Lines 197-199; 206-210; 236-2419. 

Reviewer 2 Report

The manuscript is well written, and I feel that this review is very timely. 

Just have a couple of minor suggestions:

  1. Line 21, abstract: Replace "in the last years" to "in recent years."
  2. Table 1. For the Amphiregulin substrate, the authors need to add one more in vivo reference—ADAM17 is essential for ectodomain shedding of the EGF-receptor ligand amphiregulin (PMCID: PMC5881543; DOI: 10.1002/2211-5463.12407).

Author Response

Thanks for your positive comments and suggestions to improve the quality of our manuscript.

We have changed the text according to your suggestions:

  1. Line 21, abstract: Replace "in the last years" to "in recent years." - Done.
  2. Table 1. For the Amphiregulin substrate, the authors need to add one more in vivo reference—ADAM17 is essential for ectodomain shedding of the EGF-receptor ligand amphiregulin (PMCID: PMC5881543; DOI: 10.1002/2211-5463.12407) - Done.

Round 2

Reviewer 1 Report

The authors have made adequate revisions to the text. However, as stated by the authors in their response, not much work is done using proteomics techniques, therefore there is less opportunity to discuss in tabular form. Then, I highly advise that the term "proteomics" be removed from the title of the review since it is rather deceptive. Furthermore, the authors should rearrange and state in the article that there is a significant potential that proteomics will be used to decode the TIMP-3 signalling pathway.

Author Response

We would like to thank the reviewer for his/her suggestions. We amended our manuscript accordingly, and our point-by-point responses are following:

  • I highly advise that the term "proteomics" be removed from the title of the review since it is rather deceptive.

In agreement with the reviewer, the term proteomics was removed from title

  • Furthermore, the authors should rearrange and state in the article that there is a significant potential that proteomics will be used to decode the TIMP-3 signalling pathway.

In agreement with the reviewer's suggestions we added the following to the abstract: "These methods may be used in the future to elucidate the pathological mechanisms triggered by the Sorsby’s fundus dystrophy variants of TIMP-3 or to identify proteins released by less characterized TIMP-3 target sheddases whose substrate repertoire is still limited, thus providing novel insights into the physiological and pathological functions of the inhibitor".

This concept is reiterated few times within the main text, including Intro (line 72-74) and Conclusions (823-848).